# Identifying *Early Maladaptive Schemas* from Mental Health Question Texts

§†**Sujatha Das Gollapalli**, §‡**Beng Heng Ang,**†**See-Kiong Ng**
†Institute of Data Science, National University of Singapore
‡Integrative Sciences and Engineering Programme, National University of Singapore
idssdg@nus.edu.sg,bengheng.ang@u.nus.edu,seekiong@nus.edu.sg

## Abstract

In Psychotherapy, **maladaptive schemas**– negative perceptions that *an individual has of the self, others, or the world that endure despite objective reality*– often lead to resistance to treatments and relapse of mental health issues such as depression, anxiety, panic attacks etc. Identification of early maladaptive schemas (EMS) is thus a crucial step during Schema Therapy-based counseling sessions, where patients go through a detailed and lengthy EMS questionnaire. However, such an approach is not practical in 'offline' counseling scenarios, such as community QA forums which are gaining popularity for people seeking mental health support. In this paper, we investigate both LLM (Large Language Models) and non-LLM approaches for identifying EMS labels using resources from Schema Therapy. Our evaluation indicates that recent LLMs can be effective for identifying EMS but their predictions lack explainability and are too sensitive to precise 'prompts'. Both LLM and non-LLM methods are unable to reliably address the *null* cases, i.e. cases with no EMS labels. However, we posit that the two approaches show complementary properties and together, they can be used to further devise techniques for EMS identification.

## 1 Introduction

***Background***: Psychotherapy researchers and practitioners have noted that patients with maladaptive schemas[1] do not respond fully or resist traditional cognitive-behavioral treatments and tend to relapse (Shea et al., 1990; Sanislow and McGlashan, 1998). Schema Therapy (**ST**), introduced by Young (2006), addresses this issue and builds on the widely-employed Cognitive Behavioral Therapy (Craske, 2010) and other counseling theories to conceptualize and treat mental health conditions that are due to personality disorders. **ST** has seen increasing adoption in recent counseling practice due to its effectiveness (Masley et al., 2012; Bakos et al., 2015; Taylor et al., 2017; Peeters et al., 2021).

---

**Post**: Am I going to be alone forever? I feel like I'm trying to convince myself that I'm okay when I'm not. I'm always blocking out the bad things and forgetting. I also feel like nobody cares for me and they never will. I feel truly alone.
**Expert-assigned EMS Labels**:
1. Abandonment/Instability (AB)
5. Social Isolation/Alienation (SI)

---

Table 1: A cQA forum question is shown with the EMS labels assigned by professional counselors

In Schema Therapy, *early maladaptive schemas* refer to "pervasive, unhealthy patterns of thought and behaviour" in individuals that develop from childhood experiences and affect their emotions, cognitions, relationships with others, and overall responses to life events. **ST** posits that to effectively treat mental health symptoms, it is necessary to identify these underlying maladaptive schemas and alter them with various targeted cognitive, experiential, and behavioral strategies (Young and Klosko, 1994; Young et al., 2006). In his pioneering work, Young proposed a set of 18 Early Maladaptive Schemas (**EMS**)[2] that are listed in Table 2.

***Motivation***: Community QA (cQA) forums on various topics including mental health are now ubiquitous (Griffiths et al., 2012), and studies have noted an increase in sharing of problems and seeking of both peer and professional support in cQA forums for mental health (De Choudhury and De, 2014; Prescott et al., 2017; Nobles et al., 2020). Due to the accessibility and cost effectiveness of cQA forums in providing equitable health support,

---

§Authors contributed equally.

[1]Defn. 3 at https://dictionary.apa.org/schema

[2]https://www.schematherapy.com/id73.htm

| EMS Labels Names |
| --- |
| 1. Abandonment/Instability (**AB**), 2. Mistrust/Abuse (**MA**), 3. Emotional Deprivation (**ED**) 4. Defectiveness/Shame (**DS**), 5. Social Isolation/Alienation (**SI**), 6. Dependence/Incompetance (**DI**) 7. Vulnerability to Harm or Illness (**VH**), 8. Enmeshment/Undeveloped Self (**EM**), 9. Failure to achieve (**FA**), 10. Entitlement/Grandiosity (**ET**), 11. Insufficient Self-Control/Self-Discipline (**IS**), 12. Subjugation (**SB**), 13. Self-Sacrifice (**SS**), 14. Approval-Seeking/Recognition-Seeking (**AS**), 15. Negativity/Pessimism (**NP**), 16. Emotional Inhibition (**EI**), 17. Unrelenting Standards/Hypercriticalness (**US**), 18. Punitiveness (**PU**) |
| *1. Abandonment/Instability (AB)*: The perceived instability or unreliability of those available for support and connection. Involves the sense that significant others will not be able to continue providing emotional support. connection, strength, . . . |
| *5. Social Isolation/Alienation (SI)*: The feeling that one is isolated from the rest of the world, different from other people, and/or not part of any group or community. |

Table 2: Young's list of 18 EMS labels. Due to space constraints, we only list the label names, and partial descriptions for two EMS labels and refer the reader to Table 13 in Appendix as well as the website.[2]

numerous cQA websites are now available for mental health.[3]

CQA forums invite large volumes of diverse questions. Given the limited numbers of professional counselors to cater to this demand, fast and effective "problem solving" support through automated methods for questions triage and mental health issue identification are desirable in such settings (Cohan et al., 2016; Yates et al., 2017). Against this background, the question we ask in this paper is: *Is it possible to automatically identify EMS labels characterizing negative perceptions from the texts of mental health questions?*

Consider for instance a question shown in Table 1 from the CounselChat dataset (Bertagnolli, 2020). The EMS labels assigned to this text by two trained counselors are shown along with highlighted text spans based on which they made the label assignments. We included the descriptions of the EMS labels "1. Abandonment/Instability (AB) and 5. Social Isolation/Alienation (SI)" in Table 2 for reference. We posit that effective counseling responses to such questions not only answer the overt questions but also account for the negative perceptions about oneself captured in the backstories of the inquirers. Towards this objective, we present a first study on the automatic assignment of EMS labels for counseling question texts.

***Our Contributions***: (1) We describe two techniques for predicting EMS labels through a novel application of sentence similarity and textual entailment, respectively, to Young's Schema Questionaire (YSQ), a resource employed by Schema Therapists during interactive counseling sessions. (2) In keeping with recent, exciting advancements in large language models capabilities, we employ zero-shot settings with LLMs for EMS prediction. Our goal is to examine if the sophisticated reading and language understanding abilities afforded by these models can be effectively harnessed for EMS labeling via suitable 'prompts'. (3) We provide an evaluation of our methods against LLMs on a small dataset of about thirty counseling questions annotated with EMS labels by experts. This high-quality, specialized dataset[4] contains labels and justifications for the labels provided by qualified counselors who use Schema Therapy in practice. It is a first expert-curated dataset on this topic. Though admittedly small, it provides a reasonable benchmark for a first investigation of the feasibility of automated approaches for detecting EMS labels from cQA mental health question texts, and comparing non-LLM methods with LLMs.

## 2 Methods

Schema Therapy-based counseling sessions often include the administration of a lengthy and detailed "Young's Schema Questionaire" (YSQ) as an initial step. This questionaire comprises 232 statements about oneself that are rated on

---

[3]Examples: https://www.7cups.com/qa/ and https://www.mentalhealthforum.net/forum/

[4]Our code and data is available for academic purposes at https://github.com/NUS-IDS/ems_mentalhealth.

a scale of 1 ("Completely untrue of myself")-6 ("Describes me perfectly") by the counselee.[5] For example, two statements from YSQ pertaining to the EMS labels for the question in Table 1 are

1. *I feel alienated from other people. (**SI**)*
2. *In the end, I will be alone. (**AB**)*

The counselor, who is familiar with the association of YSQ statements with the 18 EMS labels, is able to assess the potential maladaptive schemas, if any, through the scores the counselee provides in the questionaire and further confirmatory questions.

The form-filling exercise described above is not practical in the 'offline' scenario of community QA forums, requiring us to mimic this process indirectly with automated methods. Towards this goal, we harness NLP associations between the YSQ items and sentences from forum questions to infer how the YSQ form could be probably filled by the counselee. Let $\{q_1 \dots q_m\}$ refer to sentences in the input question $\mathbf{q}$, $\mathbf{Y} = \{y_1, \dots y_n\}$ represent the statements from YSQ, and $label(y_i) = E_j$ be the EMS label $E_j$ from $\{E_1, \dots E_{18}\}$ for $y_i$.

**Similarity-based Voting Predictor (SVP):** In the SVP method, we first compute the candidate set of EMS labels for a given question $\mathbf{q}$ as the set of $label(y_j)$ where $sim_{m_l}(q_i, y_j) \geq \theta$. $sim_{m_l}$ is similarity according to a sentence transformer model $m_l$ and $\theta$ is a tunable threshold. We employ multiple state-of-the-art sentence transformers $\{m_1 \dots m_m\}$ and compute our final set of EMS labels for $\mathbf{q}$ as all labels whose related statements exceed the similarity thresholds according to at least $k$ out of the available $m$ models, where the value of $k$ indicates a majority (For example, $k = 2$ when $m = 3$).

**Entailment-based Prediction model (EPM):** Natural Language Inference (NLI) models are used to compute EMS labels in EPM. Recall that the task of textual NLI involves determining whether a "hypothesis" text is true (entailment), false (contradiction), or undetermined (neutral) given a "premise" text (Poliak, 2020; Chen et al., 2021). Thus, $label(y_i)$ is included in the set of predictions if there exists $q_j \in \mathbf{q}$ that entails the hypothesis,

[5]https://psychology-training.com.au/wp-content/uploads/2017/04/Young-Schema-Questionnaire-L3.pdf

$y_i$. For the example in Table 1, the statement from YSQ, "In the end, I will be alone." is entailed by the sentence "Am I going to be alone forever?" from the question and therefore, EPM assigns the label "1. Abandonment/Instability (AB)" to this question.

Notice that EMS labels from Schema Therapy[2] are descriptive and cover a range of personal feelings, emotions, outlooks and mental thoughts. As such, with some linguistic understanding, one is able to reasonably assess if a specific EMS label is applicable to a given post using these descriptions. Since recent Large Language Models (LLMs) have shown promising abilities in language reading and understanding, we investigate if these descriptions can be effectively harnessed for predicting EMS labels with LLMs.

**Large Language Models:** Recent breakthrough research has shown that LLMs can be trained "to act in accordance with the user's intentions" and as a consequence be "prompted" to perform a range of NLP tasks (Radford et al., 2019; Brown et al., 2020; Christiano et al., 2017; OpenAI, 2023). Currently, prompt-based fine-tuning is

---

**MCPrompt**: Consider the label definitions "1. Abandonment/Instability (AB): The perceived instability . . . 2. Mistrust/Abuse: " Question: Which of the above labels are most applicable to the following context? Context: []
**YNPrompt**: Here is the definition for the label "1. Abandonment/Instability (AB): . . . Is this label applicable to the context: [] Answer Yes or No.

---

Table 3: Outline of prompts used in EMS prediction.

becoming a leading approach to solve problems that require language comprehension and in many cases shown to perform on-par with humans or previous supervised approaches (Ouyang et al., 2022; Chung et al., 2022; Touvron et al., 2023). Despite this success, LLMs are also known to suffer from shortcomings such as lack of "reliable techniques to steer the behaviour or interpret the outputs", sensitivity to prompt texts, failure to follow or detect instructions, and the ability to "make up" facts (Zhao et al., 2023; Bowman, 2023). Following the "prompt" trend, we predict EMS labels in a zero-shot setting by simply providing

| Setting | Precision↑ | Recall↑ | F1↑ | $\%O = \phi \downarrow$ | $\%G_\phi = P_\phi \uparrow$ |
|---|---|---|---|---|---|
| SVP (Threshold=0.4) | 0.276 | 0.275 | 0.261 | 31.81 | 20.00 |
| EPM (T5-large) | 0.328 | 0.307 | 0.293 | 36.36 | **40.00** |
| MCPrompt1-GPT-3.5 | 0.419 | 0.313 | 0.329 | **22.72** | 0 |
| MCPrompt5-Flan-T5 | **0.481** | 0.301 | **0.341** | **22.72** | 0 |
| YNPrompt-GPT-3.5 | 0.210 | **0.519** | 0.278 | **22.72** | 0 |
| YNPrompt-Flan-T5 | 0.380 | 0.272 | 0.283 | 54.54 | **40.00** |

Table 4: EMS Label Prediction Performance

instructions to the LLM for generating our desired output for a given question.

We explore "MCPrompt" where the LLM is required to answer a multiple-choice question requesting all applicable EMS labels for a given mental health question text given their descriptions, and the "YNPrompt" where the LLM is asked separate "Yes/No" questions on the applicability of each label for a given forum question. The outline of our prompts are shown in Table 3 with more details provided in Appendix A.2.

## 3 Results and Observations

***Dataset***: For experimental evaluation, we consider the dataset from counselchat.com that was curated by Bertagnolli (2020). This representative dataset contains about 3000 questions seeking assistance and responses from licensed therapists for various mental health topics. From the publicly-released anonymized dataset which was carefully de-identified and curated from the website by Bertagnolli (2020),[6] we considered the subset of 310 questions on the topics: anxiety, depression, trauma, self-esteem and anger-management where Schema Therapy is applicable. A random sample of about ten percent from this collection was chosen for expert annotation.

***Annotation Quality***: To ensure the *ethics, quality, and reliability* of the annotations for this specialized task, we followed previous works (Sharma et al., 2020) and selected two counselors with verified credentials and self-disclosed practical experience in Schema Therapy on Upwork.[7] The two counselors assigned EMS labels with justifications for the question texts for about five examples per hour and were paid between $40 - 50$ USD an hour, as requested. Between the two annotators, we obtained 30 questions each annotated with up to three most applicable EMS labels, on average. About 17

examples were annotated by both independently, and on this subset the Fleiss Kappa (1971) value is 0.412 indicating moderate agreement.

Recent Psychology-related studies report Kappa values indicating a wide range of agreement from low to medium depending on the complexity of the factor that is measured (Zhang et al., 2022a; Pérez et al., 2023). The moderate agreement value for our dataset sets the benchmark for EMS prediction since the annotations were made by professionally-qualified counselors who practise Schema Therapy. We also note however, the possible effect of our annotation guidelines that ask for the top-3 labels for each post. Considering the above aspects, we used the union of labels as the "ground truth" for evaluating our models. Further details on the annotation process as well as the label distribution for our dataset are included in Appendix A.4.

### 3.1 Experiments

**Models and Evaluation**: We used state-of-the-art models in our prediction algorithms available on HuggingFace (Wolf et al., 2019). For SVP and EPM, we computed pairwise similarity and entailment, respectively, between the sentences in the question and the statements in the Young's Schema Questionaire (YSQ). The top-3 transformers based on sentence similarity task performances[8] were included in SVP, whereas T5-large[9] was used for EPM. Further details on the models considered and threshold tuning are included in Appendix A.1.

For prompt-based approaches, we used the GPT-3.5 API from OpenAI[10] and Flan-T5 from Google.[11] Since EMS prediction is a multi-label classification task, we employ macro averages of the standard classification measures: precision, recall, and F1 to compare with expert (gold) anno-

---

[6]https://github.com/nbertagnolli/counsel-chat
[7]www.upwork.com

[8]https://www.sbert.net/docs/pretrained_models.html
[9]https://huggingface.co/t5-large
[10]https://openai.com/blog/openai-api
[11]https://huggingface.co/google/flan-t5-xxl

tations. Note that when there is insufficient information in the counseling question, the gold annotations set is empty and it is important for the models to correctly predict these *null* cases. We have about 19% of *null* cases in our dataset. Thus, we also measure percentage of cases when none of predictions match the gold set (overlap $O = \phi$), and the percentage of *null* cases correctly predicted by the models ($G_\phi = P_\phi$).

## 3.2 Comparison of Models

**Prediction Performance**: Table 4 lists the performance of our LLM and non-LLM approaches. After considering multiple prompts (Appendix A.2), we selected the runs with the best F1 scores. From Table 4, we see that on all standard performance measures, LLM approaches fare significantly better than the non-LLM methods. Considering that the high performance was accomplished merely by employing the label descriptions and "instructing" the LLM, it is indeed an impressive result. In particular, Flan-T5 model using MCPrompt obtains a significantly high F1 score as well as the smallest $\%O = \phi$ value compared to the other approaches. However, the MCPrompt method is not good at discovering the *null* cases. The YNPrompts provide more flexibility for handling the varying number of labels and when used with Flan-T5 also discovers a large number of the *null* cases, but the overall performance is significantly lower.

**Consistency and Explainability**: Predictions with the SVP and EPM methods have the nice property of being linked to specific statements in the Young's Schema Questionaire, a professional questionaire that is employed in practice by Schema Therapists. We were unable to obtain any explanations from the Flan-T5 models despite several attempts at prompting for the same. For GPT-3.5 runs, explanations refer to spans in the question (Example in Table 9) requiring further interpretation. In contrast, since non-LLM approaches "explain" on the basis of YSQ, this questionaire can serve as a common ground for comparing across mental health questions.

LLMs were also found to be highly sensitive to specific prompt texts (as illustrated in Appendix A.3). For minimal changes in wording and no semantic difference, the output lists of predictions can look vastly different, illustrating one of the known criticisms of prompt-approaches (Ouyang et al., 2022; Bowman, 2023).

In summary, the non-LLM and LLM approaches seem complementary in terms of prediction performance versus explainability. Considering the overall low prediction performances in this *no-data* setting and the difference in the explanatory power, we posit that the two types of approaches are best used in combination for generating "noisy" labeled data to speed up annotation efforts towards learning models with a higher degree of supervision.

## 4 Related Work

Community QA (cQA) forums are attracting significant NLP research due to their ubiquitious usage in the recent age of information sharing (Zhang et al., 2021; Sharma et al., 2020; Sosea and Caragea, 2020). Recent research works on mental health topics address symptom extraction, triage characterization as well as identification of specific conditions such as depression and self-harm behaviour (Zhang et al., 2022b; Shickel and Rashidi, 2016; Yates et al., 2017; Cohan et al., 2016; Pérez et al., 2023; Zhang et al., 2022a; Parapar et al., 2023; Nguyen et al., 2022). Unlike these previous works, we target the identification of underlying causes behind the manifested symptoms by characterizing them using EMS labels from Schema Therapy. We posit that such a theoretical grounding using domain-specific frameworks is essential for reliably interpreting model predictions in mental health applications (Fitzpatrick et al., 2017).

## 5 Conclusions

In this paper, we proposed the novel task of identifying Early Maladaptive Schemas in mental health question texts from community QA forums. To this end, we studied prediction techniques using both LLM and non-LLM approaches based on Schema Therapy, examined their strengths and weaknesses for the task, and identified their possible complementary nature regarding performance and explanations. For future, we would like to investigate if few-shot and exemplar-based approaches (Logan IV et al., 2022; Chung et al., 2022; Touvron et al., 2023) can overcome the drawbacks of LLMs highlighted in our study. We would also like to investigate recent research on incorporating human feedback through reinforcement learning (Ouyang et al., 2022) and contrastive learning to further improve prediction performance (Chen et al., 2022).

## Limitations

We proposed the novel task of identifying "maladaptive schemas" as characterized by Schema Therapy in offline counseling scenarios. In the standard scenario, this estimation is done by qualified counselors based on the YSQ forms filled by the counselee after a round of confirmatory questions (Young et al., 2006). In offline scenarios, we "impute" these values based on what is expressed in the question texts and work backwards. Thus, even with counseling expertise, since the 'client' is missing in the loop, the gold annotations are only the next best alternative. The drawbacks of using LLMs for this specific task despite their impressive prediction performance were highlighted in Section 3. For sensitive topics such as mental health, incorrect predictions can lead to wrong therapeutic interventions and outcomes. Therefore, a consistent grounding of the predictions using domain-specific frameworks (such as YSQ in this case) plays a key role in determining whether automated approaches can be adopted for final use.

## Ethics Statement

This research was conducted in accordance with the ACM Code of Ethics. The ethical considerations were fulfilled by hiring qualified counselors for annotating the mental health questions. Details on the pay, quality, and pace of annotations are included in Section 3 and Appendix A.4 and possible considerations in automating mental health prediction tasks are discussed under the Limitations section.

## Acknowledgments

This research/project is supported by the National Research Foundation, Singapore under its Industry Alignment Fund–Pre-positioning (IAF-PP) Funding Initiative and Google South & Southeast Asia Research Award 2022. Any opinions, findings and conclusions or recommendations expressed in this material are those of the author(s) and do not reflect the views of National Research Foundation, Singapore or Google LLC.

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

## A   Appendix

### A.1   Experiment Settings and List of Models

All experiments were performed on single GPU of an Nvidia Tesla cluster machine with 32GB RAM. No learning is involved, and depending on the model, predictions on the entire dataset take between a few minutes to 2 hours.

1. Three Sentence Transformers were used in the SVP model based on their performance on the sentence similarity tasks.[12] These are "all-mpnet-base-v2, all-MiniLM-L12-v2, and all-distilroberta-v1". We use 2/3 votes to determine what labels make it to the final set and tested similarity thresholds $0.3, 0.4, 0.5, 0.6$ (Table 7) for the top-5 similar statements with each EMS label.

2. We experimented with Flan-T5-XXL,[13] T5-large,[14] T5-3b,[15] and ElMo-based model from AllenNLP[16] and chose T5-large (based on the test performance) for the EPM runs.

---

Here is the definition of the label $[schema\_name]$ $[schema\_definition]$. Is this label applicable to $[context]$? Answer Yes or No.

---

Table 5: The YNPrompt used with GPT-3.5 and Flan-T5

### A.2   Prompt Experiments

The general flavor of our prompts is given in Table 3 with the precise ones used for the results in Table 4 listed in Tables 5 and 10. In the prompt string, context refers to the question text and "schema_name" and "schema_definition" were taken from https://www.schematherapy.com/id73.htm.

### A.3   LLM's Sensitivity to Prompt Texts

LLMs are known to suffer from sensitivity to the exact prompt texts, and failing to follow or detect instructions (Ouyang et al., 2022; Bowman, 2023). We observe that these drawbacks were also present for our task. For instance, for simple variations of

---

[12]https://www.sbert.net/docs/pretrained_models.html

[13]https://huggingface.co/google/flan-t5-xxl

[14]https://huggingface.co/t5-large

[15]https://huggingface.co/t5-3b

[16]https://demo.allennlp.org/

the YNPrompt string shown in Table 12, Flan-T5. Similarly the performance is quite sensitive to the precise string used for MCPrompts as can be seen in Table 11.

**Note**:   We experimented with more prompts using Flan-T5 models over GPT-3.5 due to their open-source availability. The GPT experiments were performed using the "gpt-3.5-turbo-0301" model with the temperature parameter set to 0.3.

---

Please follow these guidelines while annotating the data. For each case stated in the post in column B (counselee post), mark
1. If Schema Therapy is applicable or not (Column C is Yes/No)
2. If not applicable, provide justification in column J (Other comments)
3. If applicable, please select the top-3 most applicable EMS labels (in Columns D, F, H)

For each selected label, provide justification using either sentences in the original post (copy paste) or describe in your own words (in Columns E, G, I)
If less than 3 apply, select NONE applicable in the appropriate column.
If more than 3 apply, please list them in the last column J (Other comments)

---

Table 6: Guidelines for our annotation task

### A.4   Dataset Annotation Details

On Upwork, we posted an example question and put across our annotation task requesting for certified counselors with background knowledge and expertise in Schema Therapy. From the proposals received, we selected two counselors who were available during our data collection period and correctly assigned the labels to the example question. An excel sheet was provided with schema definitions and dropdown lists for choosing up to three or more EMS labels. The exact instructions are listed in Table 6.

We examined the annotations provided by the two annotators and removed three cases where the question texts were not about the person asking the question. For example: "My son has violent bursts of temper...". One of the annotators as well as GPT-

| Similarity Threshold | Precision↑ | Recall↑ | F1↑ | $\%O = \phi \downarrow$ | $\%G_\phi = P_\phi \uparrow$ |
|---|---|---|---|---|---|
| 0.3 | 0.251 | **0.281** | 0.249 | **22.72** | 0 |
| 0.4 | 0.276 | 0.275 | **0.261** | 31.81 | 20.00 |
| 0.5 | 0.284 | 0.243 | 0.252 | 54.54 | 60.00 |
| 0.6 | **0.324** | 0.219 | 0.240 | 77.27 | **80.00** |

Table 7: EMS Label Prediction Performance for the different similarity thresholds

(AB, 7, 25.9%), (MA, 7, 25.9%), (ED, 9, 33.33%), (DS, 8, 29.6%), (SI, 5, 18.5%), (DI, 3, 11.11%), (VH, 1, 3.7%), (EM, 2, 7.4%), (FA, 0, 0%), (ET, 0, 0%), (IS, 5, 18.5%), (SB, 5, 18.5%), (SS, 3, 11.11%), (AS, 4, 14.8%), (NP, 10, 37%), (EI, 1, 3.7%), (US, 3, 11.11%), (PU, 1, 3.7%)

Table 8: EMS Label counts and percentage spread in the annotated dataset.

3.5 provides labels for the person mentioned in the question, whereas our second annotator did not. For uniformity, these three cases were disregarded and the combined annotations from both annotators were used as our gold EMS labels for the rest of the cases. The counts for each label in this annotated dataset are shown in Table 8. The labels referring to Negativity/Pessimism, Emotional Deprivation, Abandonment/Instability, Mistrust/Abuse, and Defectiveness/Shame were noted in 25-37% of the cases seen in our dataset whereas the labels referring to "Fear of Failure" and "Entitlement" were not observed at all.

| Prompt |
|---|
| "Does the label schema_name defined as schema_definition apply to the text: [context]? Answer Yes or No. Explain your answer." |
| Yes, the label "9. FAILURE TO ACHIEVE (FA)" applies to the text about imposter syndrome in graduate school. The individual is experiencing feelings of inadequacy and fear of being exposed as a fraud, despite evidence of their success and competence.

The second question is not related to the label "9. FAILURE TO ACHIEVE (FA)" and may be better suited for a different label or category. It could potentially fall under social anxiety or avoidance behaviors. |

Table 9: Example explanation generated by GPT-3.5 for the context from Table 12 when used with the label "9. FAILURE TO ACHIEVE (FA)"

| | Prompt= Label Definition String from Table 13+ |
|---|---|
| 1 | Question: Which of the above labels (maximum 3) most apply to the following context? If no labels are applicable, just return NONE (NO). Context: [] |
| 2 | Question: Which of the above labels most apply to the following context? Select up to three. Context: [] |
| 3 | Question: Which of the above labels most apply to the following context? Select three. Context: [] |
| 4 | Question: Which of the above labels most apply to the following context? Select three. If no labels are applicable, just return NONE (NO). Context: [] |
| 5 | Question: Which of the labels above are most applicable to the following context? Select three. Context: [] |

Table 10: List of MCPrompts used in Experiments

| Setting | Precision↑ | Recall↑ | F1↑ | $\%O = \phi \downarrow$ | $\%G_\phi = P_\phi \uparrow$ |
|---|---|---|---|---|---|
| GPT-3.5 MCPrompt1 | 0.4198 | 0.3129 | 0.3297 | **22.72** | 0 |
| GPT-3.5 MCPrompt2 | 0.3457 | **0.3377** | 0.3192 | **22.72** | 0 |
| GPT-3.5 MCPrompt3 | 0.3333 | 0.3019 | 0.3000 | 27.27 | 0 |
| GPT-3.5 MCPrompt4 | 0.2962 | 0.2531 | 0.2541 | 27.27 | 0 |
| GPT-3.5 MCPrompt5 | 0.3209 | 0.3173 | 0.2973 | **22.72** | 0 |
| Flan-T5 MCPrompt1 | 0.3704 | 0.1099 | 0.1557 | 54.54 | 0 |
| Flan-T5 MCPrompt2 | **0.5185** | 0.1654 | 0.2359 | 36.36 | 0 |
| Flan-T5 MCPrompt3 | 0.4567 | 0.2117 | 0.2634 | 31.81 | 0 |
| Flan-T5 MCPrompt4 | 0.3765 | 0.2388 | 0.2641 | 36.36 | 0 |
| Flan-T5 MCPrompt5 | 0.4814 | 0.3006 | **0.3406** | **22.72** | 0 |

Table 11: EMS Label Prediction Performance for the different MCPrompts from Table 10

| |
|---|
| **Context**: How do I get over "imposter syndrome"? I'm dealing with imposter syndrome in graduate school. I know that by all accounts I am a phenomenal graduate student, and that I am well-published. I am well liked by students and faculty alike. And yet I cannot shake the feeling that I'm going to be found out as a fraud. How can I get over this feeling? |
| **Prompt** Does the label "[schema_name]" defined as "[schema_definition]" apply to the text: [context]? Answer Yes or No. **Yes output for**: 9. FAILURE TO ACHIEVE (FA) |
| **Prompt** Answer the following yes/no question using the label for [schema_name] defined [schema_definition]. Does the label apply to the text: [context]? Explain the answer. **Yes output for**: 4. DEFECTIVENESS / SHAME (DS) |
| **Prompt** Answer the following yes/no question using the label for [schema_name] defined [schema_definition]. Does the label apply to the text: [context]? **Yes output for**: 4. DEFECTIVENESS / SHAME (DS) 15. NEGATIVITY / PESSIMISM (NP) |

Table 12: Example YNPrompts with Flan-T5 illustrating the sensitivity to prompt texts and disregarding of the instruction

| Label Definitions String |
| --- |
| 1. ABANDONMENT / INSTABILITY (AB) : The belief that significant others will not be there for support and connection, leading to fear of abandonment or loss. |

1. ABANDONMENT / INSTABILITY (AB) : The belief that
significant others will not be there for support and connection, leading to fear of
abandonment or loss.
2. MISTRUST / ABUSE (MA) : The expectation that others will hurt, abuse, or betray you
in some way, leading to a general mistrust of others.
3. EMOTIONAL DEPRIVATION (ED): The belief that your emotional needs will
not be met by others, resulting in a sense of emptiness or longing for emotional connection.
4. DEFECTIVENESS / SHAME (DS) : Feeling flawed, inadequate, or unworthy, often
accompanied by a fear of being exposed as defective and a sense of shame.
5. SOCIAL ISOLATION / ALIENATION (SI) : Feeling different, disconnected, or isolated
from others, resulting in a sense of not fitting in or belonging.
6. DEPENDENCE / INCOMPETENCE (DI) : Feeling incapable of handling day-to-day
responsibilities or making decisions independently, leading to excessive reliance on others.
7. VULNERABILITY TO HARM OR ILLNESS (VH) : Constantly anticipating danger, harm, or
illness, and feeling vulnerable or unsafe in various situations.
8. ENMESHMENT / UNDEVELOPED SELF (EM) : Feeling engulfed, overshadowed, or lacking
a clear sense of self due to excessive emotional fusion with others.
9. FAILURE TO ACHIEVE (FA) : The expectation of failure or the fear of not meeting high
personal standards, leading to a persistent sense of inadequacy or disappointment.
10. ENTITLEMENT / GRANDIOSITY (ET) : A belief that you deserve special privileges,
recognition, or admiration, often accompanied by a sense of superiority or entitlement.
11. INSUFFICIENT SELF-CONTROL / SELF-DISCIPLINE (IS) : Difficulty in controlling
impulses, sticking to plans, or delaying gratification, often resulting in self-defeating behaviors.
12. SUBJUGATION (SB) : The tendency to submit to others' needs and desires while
disregarding one's own, leading to a sense of being controlled or trapped.
13. SELF-SACRIFICE (SS) : Neglecting one's own needs and prioritizing others' needs to
an excessive degree, often at the expense of personal well-being.
14. APPROVAL-SEEKING / RECOGNITION-SEEKING (AS) : An excessive need for validation,
approval, or recognition from others, leading to an overemphasis on external validation.
15. NEGATIVITY / PESSIMISM (NP) : An inclination to focus on the negative
aspects of oneself, others, or the world, leading to a pessimistic outlook and negative expectations.
16. EMOTIONAL INHIBITION (EI) : Restricting or suppressing emotions due to fear of being
overwhelmed, losing control, or being rejected or criticized.
17. UNRELENTING STANDARDS / HYPERCRITICALNESS (US) : Holding oneself to
extremely high standards of performance, often accompanied by self-criticism
and a constant sense of falling short.
18. PUNITIVENESS (PU) : The tendency to be excessively harsh or critical towards
oneself or others, often accompanied by a desire for punishment or revenge.

Table 13: The label definitions string used in MCPrompts