# OpenReview forum: "Identifying {Early Maladaptive Schemas} from Mental Health Question Texts"
_EMNLP/2023/Conference — EMNLP 2023 Findings_

### Official Review · Reviewer_ikMh · 2023-08-04

**Soundness:** 3

**Excitement:**

3: Ambivalent: It has merits (e.g., it reports state-of-the-art results, the idea is nice), but there are key weaknesses (e.g., it describes incremental work), and it can significantly benefit from another round of revision. However, I won't object to accepting it if my co-reviewers champion it.

**Paper Topic And Main Contributions:**

This paper explores novel approaches, including Large Language Models (LLMs), to predict early maladaptive schema (EMS) labels in mental health cQA forums. The paper introduces the first expert-curated dataset on this topic, which can provide a reasonable ground for investigating the feasibility of automated approaches for detecting EMS from cQA mental health question texts.

**Questions For The Authors:**

A. The exact process of computing candidate EMS labels based on sentence similarity using multiple sentence transformer models is unclear. More details would be beneficial like which similarity score was computed.
B. Can you elaborate on how entailment is determined between YSQ statements and question sentences?


**Reasons To Accept:**

The paper introduces an interesting problem of Early Maladaptive Schema Identification in mental health texts.
Evaluated various non-LLMs and LLM-based methods and offered a thorough analysis of the potential and limitations of different techniques for EMS label prediction.


**Reasons To Reject:**

The paper maintains a formal tone and technical language appropriately. However, some sections could be further polished for clarity, particularly in conveying the details of complex methodologies.  The methodology section introduces three distinct approaches logically, but it lacks detailed explanations for certain techniques. The methodology section of the paper introduces various approaches for predicting early maladaptive schemas (EMS) labels but falls short in several critical aspects.

1. The description of the Similarity-based Voting Predictor (SVP) is insufficiently detailed. The exact process of computing candidate EMS labels based on sentence similarity using multiple sentence transformer models is unclear.

2. The Entailment-based Prediction model (EPM) is introduced briefly but lacks clarity in its application. The section does not elaborate on how entailment is determined between YSQ statements and question sentences.


**Reproducibility:**

N/A: Doesn't apply, since the paper does not include empirical results.

**Reviewer Confidence:**

4: Quite sure. I tried to check the important points carefully. It's unlikely, though conceivable, that I missed something that should affect my ratings.

---

> ### Author Rebuttal · Authors · 2023-08-29
>
> We apologize that some clarity was lost due to space constraints of short paper. Further details on these two methods are included in Appendix A.1.
>
> 1. We computed pairwise similarity (or alternatively, entailment) between the sentences in the question and the statements in the Young’s Schema Questionaire (YSQ) resource available from Schema Therapy. The YSQ statements are grouped into EMS categories/labels.
>
> For SVP, we applied three state-of-the-art sentence transformer models to obtain sentence embeddings and compute cosine similarity. An EMS label (attached to the corresponding YSQ statement) is only applied if at least two of the transformers provide a similarity score above a threshold. The effect of these thresholds are illustrated in Table-7 of the appendix.
>
> 2. For the EPM method, we used an off-the-shelf T5-large entailment prediction model
> (T5 is already fine-tuned on the entailment task: JMLR 2020 "Exploring the Limits of Transfer Learning with a Unified
> Text-to-Text Transformer"). We choose EMS labels corresponding to the YSQ statement for which entailment holds. For example (lines 161-163), the statement in YSQ "In the end, I will be alone" is entailed by the sentence in the post from Table-1 "Am I going to be alone forever?" according to the T5 model prediction.

---

### Official Review · Reviewer_14Gv · 2023-08-13

**Typos Grammar Style And Presentation Improvements:** In line 162, the statement from YSQ s…
**Soundness:** 3

**Excitement:**

2: Mediocre: This paper makes marginal contributions (vs non-contemporaneous work), so I would rather not see it in the conference.

**Paper Topic And Main Contributions:**

This paper is about mental health analysis. The paper proposes a task of identifying Early Maladaptive Schemas in mental health question texts from community QA forums. In addition, this paper tests the performance of LLMs and non-LLMs methods on the task. There are some findings about EMS detection.

**Reasons To Accept:**

>Describing two methods for predicting EMS labels through a novel application of sentence similarity and textual entailment.
>Providing an evaluation of the methods using LLMs and non-LLMs
>Constructing a small EMS dataset.

**Reasons To Reject:**

>The overall novelty of this paper is limited. The proposed SVP method exists. In addition, the EMS prediction task is similar to depressive symptoms detection tasks. Therefore, there are many related methods leveraging semantic information, such as:
Zhang, Z., Chen, S., Wu, M., & Zhu, K. Q. (2022a). Psychiatric scale guided risky post screening for early detection of depression. arXiv preprint arXiv:2205.09497. Zhang, Z., Chen, S., Wu, M., & Zhu, K. (2022b). Symptom identification for interpretable detection of multiple mental disorders on social media. In Proceedings of the 2022 conference on empirical methods in natural language processing (pp. 9970–9985).
In addition, although the LLMs' method and evaluation are useful for the EMS prediction task, there are no interesting findings, and the results can not help justify the lack of interpretability. The authors could add a chain of thought to learn more about the performance of LLMs.

>Some contents need to be included and need to be more clearly described: The specific structure and usage of the EPM method.  Which F1 (micro, macro, average) has been used?  The label distribution of the dataset (imbalanced or balanced)?

>Some experiments should be added, such as ablation studies about non-LLMs-based models.

>The Fleiss Kappa is low. What is the reason, are two annotators enough, and is the guideline set reasonable?


**Reproducibility:**

3: Could reproduce the results with some difficulty. The settings of parameters are underspecified or subjectively determined; the training/evaluation data are not widely available.

**Reviewer Confidence:**

4: Quite sure. I tried to check the important points carefully. It's unlikely, though conceivable, that I missed something that should affect my ratings.

---

> ### Author Rebuttal · Authors · 2023-08-29
>
> 1. We respectfully disagree with the reviewer's comment on novelty. The proposed task is different from
> existing works on depression including symptom extraction and screening/triage works. Borrowing from the clinical field, one way to illustrate this point is the difference between cause, symptom, and condition/disease. Schema Therapy from Psychology/Counseling provides a theoretical framework for understanding underlying causes behind manifested symptoms and is used in practical counseling for a range of mental health conditions such as depression and bipolar disorder.
>
> As such, the novelty of this paper pertains not to the SVP alone but a study on the novel application of Schema Therapy (unseen in studies so far) via the
> EMS prediction task. We study this problem in a zero-shot setting using sentence similarity (SVP), entailment, as well as
> LLMs which are being harnessed for their high-level of comprehension on a variety of tasks.
>
> We are happy to highlight this aspect in related work comparison and include the references suggested by the reviewer in related works.
>
> 2. In this first study for a short paper, we focused on the zero-shot setting for comparing LLMs with non-LLM approaches. Further details on EPM as well as annotation are included in the Appendix (A.1 and A.4)  due to space constraints. We will include the other finer details requested by the reviewer.
>
> 3.Previous Psychology-related studies report Kappa values indicating a wide range of agreement from low to medium depending on the complexity of the factor that is measured (For example, https://arxiv.org/abs/2308.10758 and https://aclanthology.org/2022.emnlp-main.677.pdf). In our work, we are the first to report the agreement values of predicting EMS by two professional counselors and this sets the benchmark for our task. The moderate agreement between annotators (as indicated by Fleiss Kappa), is also possibly an artifact of our annotation guidelines task where we requested for top-3 labels for a post from each annotator. As such, both the annotators are qualified and practicing counselors who apply a label and provide explanations for each label by highlighting the appropriate spans in the questions. Considering this aspect, the union of labels was used as ground truth. Please find further details in Appendix A.4.
>
> Typo: We are not sure what this comment means.

---

### Official Review · Reviewer_Gb3g · 2023-08-13

**Soundness:** 3

**Excitement:**

2: Mediocre: This paper makes marginal contributions (vs non-contemporaneous work), so I would rather not see it in the conference.

**Paper Topic And Main Contributions:**

This work is a first attempt at understanding a concept called early maladaptive schemas from community question answering forums. The approach involves automating filling up of Young's schema questionnaire using various methods (both LLM and non-LLM based) to obtain labels of schemas (18 classes) for each post.

**Reasons To Accept:**

* A new idea to apply a concept from schema theory in psychology to NLP texts- the work can be beneficial to other clinical NLP researchers and be impactful for studying triage.
* The paper is well-written and easy to understand.

**Reasons To Reject:**

* Many obvious experiments come to mind given its a first attempt at understanding the concept in language domain -- for example, the models could be finetuned for the methods, or a model trained on mental health datasets specifically could be used. Studies could be done across multiple models to check for inherent knowledge of identifying EMS. The choice of models is not well-motivated either.
* Consistency and Explainability: although its a section, the discussion on non-LLM is limited to one line and doesn't have any specific experiments designed for either consistency or explainability.
* The work is small and limited in its applications, I would like to see more experiments and settings for a thorough evaluation. The work in its current form might be better suited for a related workshop.

**Reproducibility:**

3: Could reproduce the results with some difficulty. The settings of parameters are underspecified or subjectively determined; the training/evaluation data are not widely available.

**Reviewer Confidence:**

4: Quite sure. I tried to check the important points carefully. It's unlikely, though conceivable, that I missed something that should affect my ratings.

**Typos Grammar Style And Presentation Improvements:**

* Please acknowledge the possible impacts of predicting EMS labels in the Ethics section.
* Table 3-- why is 0.3006 bolded? It might also be better to have results upto 3 decimal places since the point holds with less decimals.

---

> ### Author Rebuttal · Authors · 2023-08-29
>
> 1. Existing mental health datasets do not include EMS labels and we are the first to propose this novel task. Given the lack of annotated data, we presented our study in the zero-shot setting where no learning and fine-tuning is employed and the annotated data collected by us has been used for evaluation alone (of non-LLM versus LLM approaches).
>
> 2. We apologize for the confusion on the use of “consistency and explainability”. We are referring to the fact that LLMs provide explanations for each case referring to spans in the text making it impossible to compare across cases. In contrast, non-LLM approaches “explain” on the basis of the Young’s Schema Questionaire, so this is common ground using which two cases can be compared.
>
> 3. Ours is the first study that grounds mental health predictions using Schema Therapy, that is practically employed in real-world counseling, will provide the theoretical grounding for development of further mental health applications. We explored the viability of LLMs and non-LLM approaches for EMS labels from Schema Therapy in a zero-shot setting in this paper and find that there is large room for improvement and future work.
>
> We thank the reviewer for catching the presentation problem on bolding and noted the comment on three decimals. We will also add the problems highlighted in limitations to the Ethics statement as suggested.

---

### Official Review · Reviewer_CaET · 2023-08-13

**Soundness:** 4

**Excitement:**

3: Ambivalent: It has merits (e.g., it reports state-of-the-art results, the idea is nice), but there are key weaknesses (e.g., it describes incremental work), and it can significantly benefit from another round of revision. However, I won't object to accepting it if my co-reviewers champion it.

**Paper Topic And Main Contributions:**

The paper discusses the task of identifying Early Maladaptive Schemas (EMS) in the mental health question texts which are collected from a community QA forum. Two different approaches are also discussed and compared in the paper with varying performance across the dataset.

**Questions For The Authors:**

Please refer to the weaknesses of the paper above, especially 2, 3, 4. What was the reason for not performing any level of fine-tuning on the dataset.

**Reasons To Accept:**

1. The paper highlights a significant application of natural language processing in the realm of mental health.
2. The authors have curated a valuable dataset for identifying EMS from patient texts or conversations.

**Reasons To Reject:**

1. The paper's dataset presentation lacks clarity, making it challenging to follow without specific examples.
2. The discussion of dataset de-identification, critical for dataset release, is absent.
3. The inter-annotator agreement is notably low. There's a lack of discussion on the adjudication strategy and revisions that shaped the final test dataset.
4. Incorporating few-shot learning, especially for null labels, and implementing contrastive learning could enhance the results. Consider making the dataset publicly available, as it could be a valuable resource for the community.

I kindly request the authors to provide a more detailed discussion of the dataset and consider its public availability for the benefit of the community.

**Reproducibility:**

4: Could mostly reproduce the results, but there may be some variation because of sample variance or minor variations in their interpretation of the protocol or method.

**Reviewer Confidence:**

4: Quite sure. I tried to check the important points carefully. It's unlikely, though conceivable, that I missed something that should affect my ratings.

---

> ### Author Rebuttal · Authors · 2023-08-29
>
> 1. We apologize that more examples could not be squeezed in due to lack of space.  In addition to the example in the Introduction, one more example is included in the Appendix.
>
> 2. We used a subset of the publicly-available dataset from counselchat.com provided by (Nicolas Bertagnolli. 2020. Counsel chat: Bootstrapping high-quality therapy data. https://github.com/nbertagnolli/counsel-chat) for our study. The referenced paper details the de-identification and curation process adopted for releasing this dataset publicly. We will highlight this specifically in our paper.
>
> 3. Previous Psychology-related studies report Kappa values indicating a wide range of agreement from low to medium depending on the complexity of the factor that is measured (For example, https://arxiv.org/abs/2308.10758 and https://aclanthology.org/2022.emnlp-main.677.pdf). In our work, we are the first to report the agreement values of predicting EMS by two professional counselors and this sets the benchmark for our task. The moderate agreement between annotators (as indicated by Fleiss Kappa), is also possibly an artifact of our annotation guidelines task where we requested for top-3 labels for a post from each annotator. As such, both the annotators are qualified and practicing counselors who apply a label and provide explanations for each label by highlighting the appropriate spans in the questions. Considering this aspect, the union of labels was used as ground truth. Please find further details in Appendix A.4.
>
> 4. The reviewer’s comment on few-shot learning as well as contrastive learning is well-noted for further study. The original dataset as well as our subset of annotations will also be publicly available as mentioned in Footnote 5.

---

### Meta-Review · Area_Chair_trZ5 · 2023-09-19

**Recommendation:** 3

**Metareview:**

The paper explores the task of identifying Early Maladaptive Schemas (EMS) in mental health texts obtained from community question-answering forums.  While the work makes valuable contributions, particularly in dataset curation and initial analysis, it also suffers from a series of shortcomings that affect its overall impact. All reviewers acknowledge the significance of applying NLP in the mental health domain. he dataset generated is of noteworthy value, especially since it focuses on Early Maladaptive Schemas, a topic not previously covered in available resources. Multiple reviewers noted that the dataset presentation could be more precise, with more context and examples for easier comprehension. Given the sensitive nature of mental health data, reviewers have expressed concerns over the lack of discussion regarding dataset de-identification. The reviewers found the paper’s contributions to be incremental. They recommend further experiments for a more thorough evaluation. Given the strengths and weaknesses pointed out by the reviewers, the paper is recommended accepted into the Findings. The authors are encouraged to consider the detailed feedback from the reviewers to enhance the quality and impact of their work.

---

### Decision · Program_Chairs · 2023-10-07

**Decision:**

Accept-Findings

**Comment:**

The paper explores the task of identifying Early Maladaptive Schemas (EMS) in mental health texts obtained from community question-answering forums.  While the work makes valuable contributions, particularly in dataset curation and initial analysis, it also suffers from a series of shortcomings that affect its overall impact. All reviewers acknowledge the significance of applying NLP in the mental health domain. he dataset generated is of noteworthy value, especially since it focuses on Early Maladaptive Schemas, a topic not previously covered in available resources. Multiple reviewers noted that the dataset presentation could be more precise, with more context and examples for easier comprehension. Given the sensitive nature of mental health data, reviewers have expressed concerns over the lack of discussion regarding dataset de-identification. The reviewers found the paper’s contributions to be incremental. They recommend further experiments for a more thorough evaluation. Given the strengths and weaknesses pointed out by the reviewers, the paper is recommended accepted into the Findings. The authors are encouraged to consider the detailed feedback from the reviewers to enhance the quality and impact of their work.